# Small batch deep reinforcement learning

**Johan Obando-Ceron** [*]
Mila, Université de Montréal
jobando0730@gmail.com

**Marc G. Bellemare**
Mila, Université de Montréal
bellemam@mila.quebec

**Pablo Samuel Castro**
Google DeepMind
Mila, Université de Montréal
psc@google.com

## Abstract

In value-based deep reinforcement learning with replay memories, the batch size parameter specifies how many transitions to sample for each gradient update. Although critical to the learning process, this value is typically not adjusted when proposing new algorithms. In this work we present a broad empirical study that suggests *reducing* the batch size can result in a number of significant performance gains; this is surprising, as the general tendency when training neural networks is towards larger batch sizes for improved performance. We complement our experimental findings with a set of empirical analyses towards better understanding this phenomenon.

## 1 Introduction

One of the central concerns for deep reinforcement learning (RL) is how to efficiently make the most use of the collected data for policy improvement. This is particularly important in online settings, where RL agents learn while interacting with an environment, as interactions can be expensive. Since the introduction of DQN [Mnih et al., 2015], one of the core components of most modern deep RL algorithms is the use of a finite *replay memory* where experienced transitions are stored. During learning, the agent samples mini-batches from this memory to update its network parameters.

Since the policy used to collect transitions is changing throughout learning, the replay memory contains data coming from a mixture of policies (that differ from the agent's current policy), and results in what is known as *off-policy* learning. In contrast with training data for supervised learning problems, online RL data is highly *non-stationary*. Still, at any point during training the replay memory exhibits a distribution over transitions, which the agent samples from at each learning step. The number of sampled transitions at each learning step is known as the *batch size*, and is meant to produce an unbiased estimator of the underlying data distribution. Thus, in theory, larger batch sizes should be more accurate representations of the true distribution.

Some in the supervised learning community suggest that learning with large batch sizes leads to better optimization [Shallue et al., 2019], since smaller batches yield noisier gradient estimations. Contrastingly, others have observed that larger batch sizes tend to converge to "sharper" optimization landscapes, which can result in worsened generalization [Keskar et al., 2017]; smaller batches, on the other hand, seem to result in "flatter" landscapes, resulting in better generalization.

Learning dynamics in deep RL are drastically different than those observed in supervised learning, in large part due to the data non-stationarity mentioned above. Given that the choice of batch size will have a direct influence on the agent's sample efficiency and ultimate performance, developing a better understanding of its impact is critical. Surprisingly, to the best of our knowledge there have been no studies exploring the impact of the choice of batch size in deep RL. Most recent works have focused on related questions, such as the number of gradient updates per environment step [Nikishin et al., 2022, D'Oro et al., 2023, Sokar et al., 2023], but have kept the batch size fixed.

---

[*]Work done during an internship at Google DeepMind

37th Conference on Neural Information Processing Systems (NeurIPS 2023).

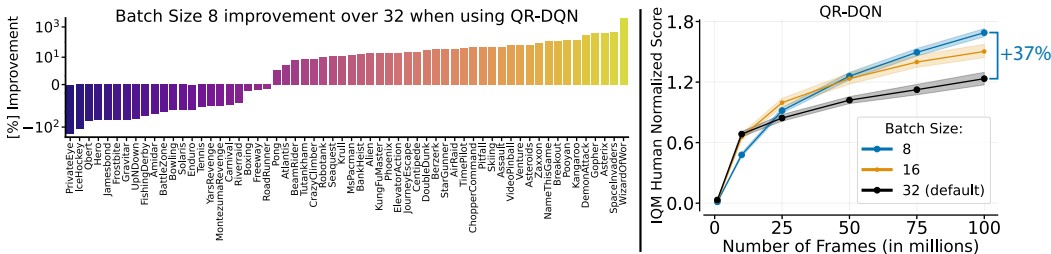

Figure 1: Evaluating QR-DQN [Dabney et al., 2018a] with varying batch sizes over all 60 Atari 2600 games. **(Left)** Average improvement obtained when using a batch size of 8 over 32 (default); **(Right)** Aggregate Interquantile Mean [Agarwal et al., 2021] of human normalized scores. All games run for 3 seeds, with shaded areas displaying 95% stratified bootstrap confidence intervals.

In this work we conduct a broad empirical study of batch size in online value-based deep reinforcement learning. We uncover the surprising finding that *reducing* the batch size seems to provide substantial performance benefits and computational savings. We showcase this finding in a variety of agents and training regimes (section 3), and conduct in-depth analyses of the possible causes (section 4). The impact of our findings and analyses go beyond the choice of the batch size hyper-parameter, and help us develop a better understanding of the learning dynamics in online deep RL.

## 2    Background

A reinforcement learning problem is typically formulated as a Markov decision process (MDP), which consists of a 5-tuple $\langle \mathcal{S}, \mathcal{A}, \mathcal{P}, \mathcal{R}, \gamma, \rangle$, where $\mathcal{S}$ denotes the state space, $\mathcal{A}$ denotes the actions, $\mathcal{P} : \mathcal{S} \times \mathcal{A} \to Dist(\mathcal{S})$ encodes the transition dynamics, $\mathcal{R} : \mathcal{S} \times \mathcal{A} \to \mathbb{R}$ is the reward function, and $\gamma \in [0, 1)$ is a discount factor. The aim is to learn a *policy* $\pi_\theta : \mathcal{S} \mapsto \mathcal{A}$ parameterized by $\theta$ such that the sum of discounted returns $\mathbb{E}_{\pi_\theta} \left[ \sum_{t=1}^{\infty} \gamma^t r_t \right]$ is maximized; here, the state-action trajectory $(\mathbf{s}_0, \mathbf{a}_0, \mathbf{s}_1, \mathbf{a}_1, \ldots)$ is obtained by sampling an action $\mathbf{a}_t \sim \pi_\theta (\cdot \mid \mathbf{s}_t)$ and reaching state $\mathbf{s}_{t+1} \sim \mathcal{P} (\cdot \mid \mathbf{s}_t, \mathbf{a}_t)$ at each decision step $t$, and $r_t \sim \mathcal{R} (\cdot \mid \mathbf{s}_t, \mathbf{a}_t)$.

In value-based methods, the policy is obtained as the argmax of a learned $Q$-function: $\pi_\theta(s) \equiv \arg\max_{a \in \mathcal{A}} Q_\theta(s, a)$. This function aims to approximate the optimal state-action values $Q^*$, defined via the well-known Bellman recurrence: $Q^*(\mathbf{s}_t, \mathbf{a}_t) = \max_{\mathbf{a}'} \mathbb{E}[\mathcal{R}(\mathbf{s}_t, \mathbf{a}_t) + \gamma Q^* (\mathbf{s}_{t+1}, \mathbf{a}_{t+1})]$, and is typically learned using $Q$-learning [Watkins and Dayan, 1992, Sutton and Barto, 2018].

To deal with large state spaces, such as all possible images in an Atari 2600 game, Mnih et al. [2015] introduced DQN, which combined Q-learning with deep neural networks to represent $Q_\theta$. A large *replay buffer* $D$ is maintained to store experienced transitions, from which mini-batches are sampled to perform learning updates [Lin, 1992]. Specifically, *temporal difference learning* is used to update the network parameters with the following loss function: $L(\theta) = \mathbb{E}_{(s_t, a_t, r_t, s_{t+1}) \sim D}[((r_t + \gamma \max_{a' \in \mathcal{A}} Q_{\bar{\theta}}(s_{t+1}, a_{t+1})) - Q_\theta(s_t, a_t))^2]$. Here $Q_{\bar{\theta}}$ is a *target network* that is a delayed copy of $Q_\theta$, with the parameters synced with $Q_\theta$ less frequently than $Q_\theta$ is updated.

Since the introduction of DQN, there have been a number of algorithmic advances in deep RL agents, in particular those which make use of distributional RL [Bellemare et al., 2017], introduced with the C51 algorithm. The Rainbow agent combined C51 with other advances such as multi-step learning and prioritized replay sampling [Hessel et al., 2018]. Different ways of parameterizing return distributions were proposed in the form of the IQN [Dabney et al., 2018b] and QR-DQN [Dabney et al., 2018a] algorithms. For reasons which will be clarified below, most of our evaluations and analyses were conducted with the QR-DQN agent.

## 3    The small batch effect on agent performance

In this section we showcase the performance gains that arise when training with smaller batch sizes. We do so first with four standard value-based agents (§3.1), with varying architectures (§3.2), agents optimized for sample efficiency (§3.3), and with extended training (§3.4). Additionally, we explore the impact of reduced batch sizes on exploration (§3.5) and computational cost (§3.6).

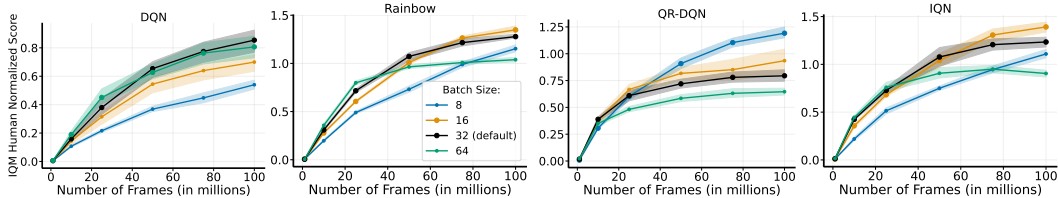

Figure 3: IQM for human normalized scores for DQN, Rainbow, QR-DQN, and IQN over 20 games. All games run with 3 independent seeds, shaded areas representing 95% confidence intervals.

**Experimental setup:** We use the Jax implementations of RL agents, with their default hyper-parameter values, provided by the Dopamine library [Castro et al., 2018][2] and applied to the Arcade Learning Environment (ALE) [Bellemare et al., 2013].[3] It is worth noting that the default batch size is 32, which we indicate with a **black** color in all the plots below, for clarity. We evaluate our agents on 20 games chosen by Fedus et al. [2020] for their analysis of replay ratios, picked to offer a diversity of difficulty and dynamics. To reduce the computational burden, we ran most of our experiments for 100 million frames (as opposed to the standard 200 million). For evaluation, we follow the guidelines of Agarwal et al. [2021]. Specifically, we run 3 independent seeds for each experiment and report the human-normalized *interquantile mean (IQM)*, aggregated over the 20 games, configurations, and seeds, with the 95% stratified bootstrap confidence intervals. Note that this means that for most of the aggregate results presented here, we are reporting mean and confidence intervals over 60 independent seeds. All experiments were run on NVIDIA Tesla P100 GPUs.

## 3.1 Standard agents

We begin by investigating the impact reducing the batch size can have on four popular value-based agents, which were initially benchmarked on the ALE suite: DQN [Mnih et al., 2015], Rainbow [Hessel et al., 2018] (Note that Dopamine uses a "compact" version of the original Rainbow agent, including only multi-step updates, prioritized replay, and C51), QR-DQN [Dabney et al., 2018a], and IQN [Dabney et al., 2018b]. In Figure 3 we can observe that, in general, reduced batch size results in improved performance. The notable exception is DQN, for which we provide an analysis and explanation for why this is the case below. To verify that our results are not a consequence of the set of 20 games used in our analyses, we ran QR-DQN (where the effect is most observed) over the full

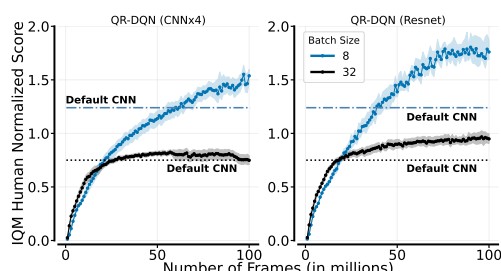

Figure 2: IQM for human normalized scores with varying neural network architectures over 20 games, with 3 seeds per experiment. Shaded areas represent 95% stratified bootstrap confidence intervals.

60 games in the suite and report the results in Figure 19. Remarkably, a batch size of 8 results in significant gains on 38 out of the full 60 games, for an average performance improvement of 98.25%.

## 3.2 Varying architectures

Although the CNN architecture originally introduced by DQN [Mnih et al., 2015] has been the backbone for most deep RL networks, there have been some recent works exploring the effects of varying architectures [Espeholt et al., 2018, Agarwal et al., 2022, Sokar et al., 2023]. We investigate the small batch effect by varying the QR-DQN architecture in two ways: **(1)** expanding the convolutional widths by 4 times (resulting in a substantial increase in the number of parameters), and **(2)** using the Resnet architecture proposed by Espeholt et al. [2018] (which results in a similar number of parameters to the original CNN architecture, but is a deeper network). In Figure 2 we can observe that not only do reduced batch sizes yield improved performance, but they are better able to leverage the increased number of parameters (CNNx4) and the increased depth (Resnet).

---

[2]Dopamine code available at https://github.com/google/dopamine.
[3]Dopamine uses sticky actions by default [Machado et al., 2018].

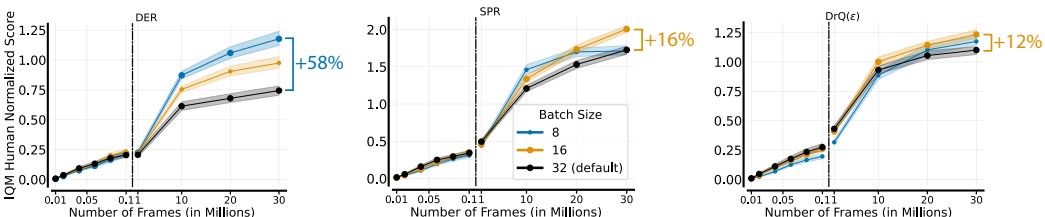

Figure 4: Measured IQM of human-normalized scores on the 26 100k benchmark games, with varying batch sizes, of DER, SPR, and DrQ($\epsilon$). We evaluate performance at 100k agent steps (or 400k environment frames), and at 30 million environment frames, run with 6 independent seeds for each experiment, and shaded areas display 95% confidence intervals.

## 3.3 Atari 100k agents

There has been an increased interest in evaluating Atari agents on very few environment interactions, for which Kaiser et al. [2020] proposed the 100k benchmark[4]. We evaluate the effect of reduced batch size on three of the most widely used agents for this regime: Data-efficient Rainbow (DER), a version of the Rainbow algorithm with hyper-parameters tuned for faster early learning [van Hasselt et al., 2019]; DrQ($\epsilon$), which is a variant of DQN that uses data augmentation [Agarwal et al., 2021]; and SPR, which incorporates self-supervised learning to improve sample efficiency [Schwarzer et al., 2020]. For this evaluation we evaluate on the standard 26 games for this benchmark [Kaiser et al., 2020], aggregated over 6 independent trials.

In Figure 4 we include results both at the 100k benchmark (left side of plots), and when trained for 30 million frames. Our intent is to evaluate the batch size effect on agents that were optimized for a different training regime. We can see that although there is little difference in 100k, there is a much more pronounced effect when trained for longer. This finding suggests that reduced batch sizes enables continued performance improvements when trained for longer.

## 3.4 Training Stability

To further investigate whether reduced batch sizes enables continual improvements with longer training, we extend the training of QR-DQN up to the standard 200 million frames. In Figure 5 we can see that training performance tends to plateau for the higher batch sizes. In contrast, the smaller batch sizes seem to be able to continuously improve their performance.

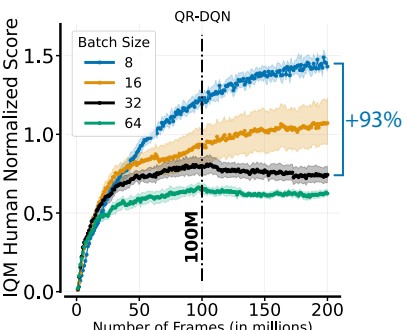

Figure 5: Measuring IQM for human-normalized scores when training for 200 million frames. Results aggregated over 20 games, where each experiment was run with 3 independent seeds and we report 95% confidence intervals.

## 3.5 Impact on exploration

The simplest and most widely used approach for exploration is to select actions randomly with a probability $\epsilon$, as opposed to selecting them greedily from the current $Q_\theta$ estimate. The increased variance resulting from reduced batch sizes (as we will explore in more depth below) may also result in a natural form of exploration. To investigate this, we set the target $\epsilon$ value to 0.0 for QR-DQN[5]. In Figure 6 we compare performance across four known hard exploration games [Bellemare et al., 2016, Taiga et al., 2020] and observe that reduced batch sizes tends to result in improved performance for these games.

---

[4]Here, 100k refers to agent steps, or 400k environment frames, due to skipping frames in the standard training setup.

[5]Note that we follow the training schedule of Mnih et al. [2015] where the $\epsilon$ value begins at 1.0 and is linearly decayed to its target value over the first million environment frames.

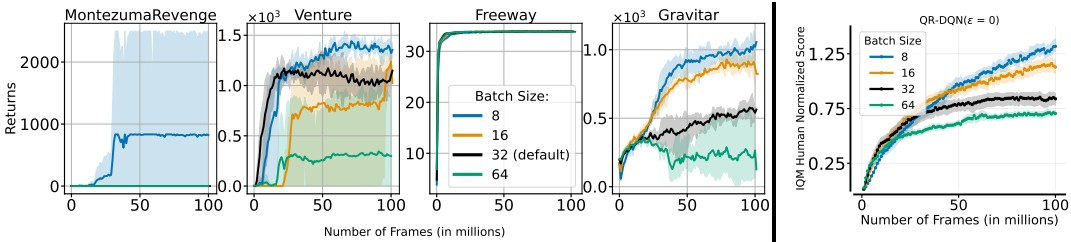

Figure 6: **Left:** Performance of QR-DQN on four hard exploration games with a target $\epsilon$ value of $0.0$, and with varying batch sizes. **Right:** Aggregate IQM of human-normalized scores over 20 games with a target $\epsilon$ value of $0.0$. In all the plots 3 independent seeds were used for each game/batch-size configuration, with shaded areas representing 95% confidence intervals.

Many methods have been proposed to address the exploitation-exploration dilemma, and some techniques emphasize exploration by adding noise directly to the parameter space of agents [Fortunato et al., 2018, Plappert et al., 2018, Hao et al., 2023, Eberhard et al., 2023], which inherently adds variance to the learning process. Our analyses show that increasing variance by reducing the batch size may result in similar beneficial exploratory effects, as the mentioned works suggest.

### 3.6 Computational impact

Empirical advances in deep reinforcement learning are generally measured with respect to sample efficiency; that is, the number of environment interactions required before achieving a certain level of performance. It fails to capture computational differences between algorithms. If two algorithms have the same performance with respect to environment interactions, but one takes twice as long to perform each training step, one would clearly opt for the faster of the two. This important distinction, however, is largely overlooked in the standard evaluation methodologies used by the DRL community.

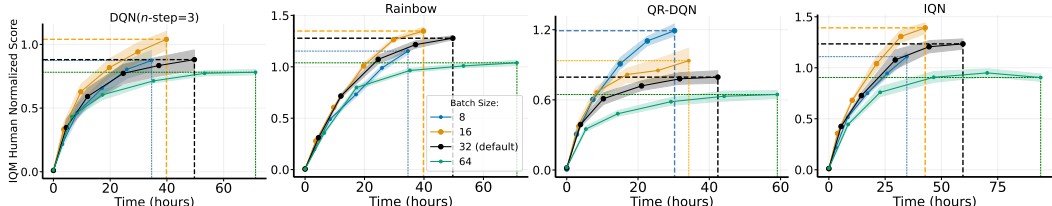

Figure 7: Measuring wall-time versus IQM of human-normalized scores when varying batch sizes in DQN (with $n$-step set to 3), Rainbow, QR-DQN, and IQN over 20 games. Each experiment had 3 independent runs, and the confidence intervals show 95% confidence intervals.

We have already demonstrated the performance benefits obtained when reducing batch size, but an additional important consequence is the reduction in computation wall-time. Figure 7 demonstrates that not only can we obtain better performance with a reduced batch size, but we can do so at a fraction of the runtime. As a concrete example, when changing the batch size of QR-DQN from the default value of 32 to 8, we achieve both a 50% performance increase and a 29% speedup in wall-time. It may seem surprising that smaller batch sizes have a faster runtime, since larger batches presumably make better use of GPU parallelism. However, as pointed out by Masters and Luschi [2018], the speedups may be a result of a smaller memory footprint, enabling better machine throughput.

Considering the unsuitable increase in computational requirements, progress with deep learning demands more compute-efficient training methods. A natural direction is to eliminate algorithmic inefficiencies in the learning process, aiming to reduce time, energy consumption and carbon footprint associated with training these models [Bartoldson et al., 2023, Chen et al., 2021]. Figure 14 illustrates the wall-time reduction when using high-capacity neural networks and smaller batch size value. This motivates a fundamental trade-off in the choice of batch size, and the way of how we benchmark deep reinforcement learning algorithms.

**Key observations on reduced batch sizes:**

- They generally improve performance, as evaluated across a variety of agents and network architectures.
- When trained for longer, the performance gains continue, rather than plateauing.
- They seem to have a beneficial effect on exploration.
- They result in faster training, as measured by wall-time.

## 4 Understanding the small batch effect

Having demonstrated the performance benefits arising from a reduced batch size across a wide range of tasks, in this section we seek to gain some insight into possible causes. We will focus on QR-DQN, as this is the agent where the small batch effect is most pronounced (Figure 3). We begin by investigating possible confounding factors for the small batch effect, and then provide analyses on the effect of reduced batch sizes on network dynamics.

### 4.1 Relation to other hyperparameters

**Learning rates** It is natural to wonder whether an improved learning rate could produce the same effect as simply reducing the batch size. In Figure 8 we explored a variety of different learning rates and observe that, although performance is relatively stable with a batch size of 32, it is unable to reach the performance gains obtained with a batch size of 8 or 16. Figure 8 shows that the smaller the learning rate, the larger batch size needs to be, and thus the longer training takes. This result aligns well with the findings of Wilson and Martinez [2003].

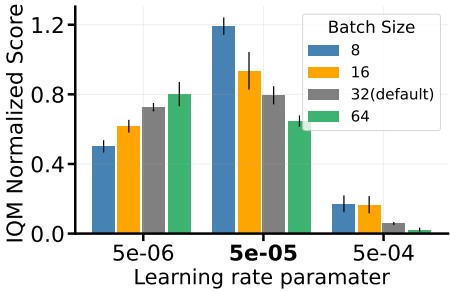

Figure 8: Varying batch sizes for different learning values. Results aggregated IQM of human-normalized scores over 20 games for QR-DQN.

**Second order optimizer effects** All our experiments, like most modern RL agents, use the Adam optimizer [Kingma and Ba, 2015], a variant of stochastic gradient descent (SGD) that adapts its learning rate based on the first- and second-order moments of the gradients, as estimated from mini-batches used for training. It is thus possible that smaller batch sizes have a second-order effect on the learning-rate adaptation that benefits agent performance. To investigate this we evaluated, for each training step, performing multiple gradient updates on subsets of the original sampled batch; we define the parameter $BatchDivisor$ as the number of gradient updates and dividing factor (where a value of 1 is the default setting). Thus, for a $BatchDivisor$ of 4, we would perform 4 gradient updates with subsets of size 8 instead of a single gradient update with a mini-batch of size 32. With an optimizer like SGD this has no effect (as they are mathematically equivalent), but we may see differing performance due to Adam's adaptive learning rates. Figure 9 demonstrates that, while there are differences, these are not consistent nor significant enough to explain the performance boost.

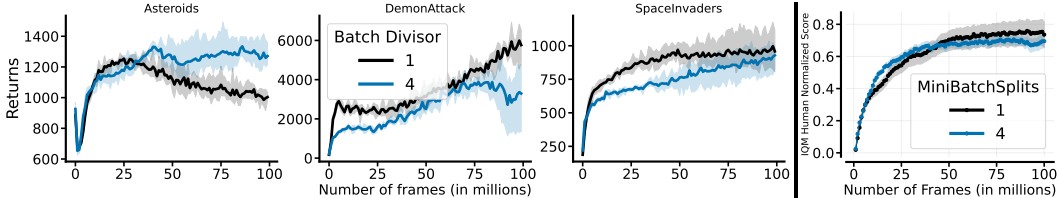

Figure 9: Varying the number of gradient updates per training step, for a fixed batch size of 32. **Left:** Performance of QR-DQN on three games with different $BatchDivisor$ value. **Right:** Results aggregated IQM of human-normalized scores over 20 games for QR-DQN.

**Relationship with multi-step learning** In Figure 3 we observed that DQN was the only agent where reducing batch size did not improve performance. Recalling that the Dopamine version of

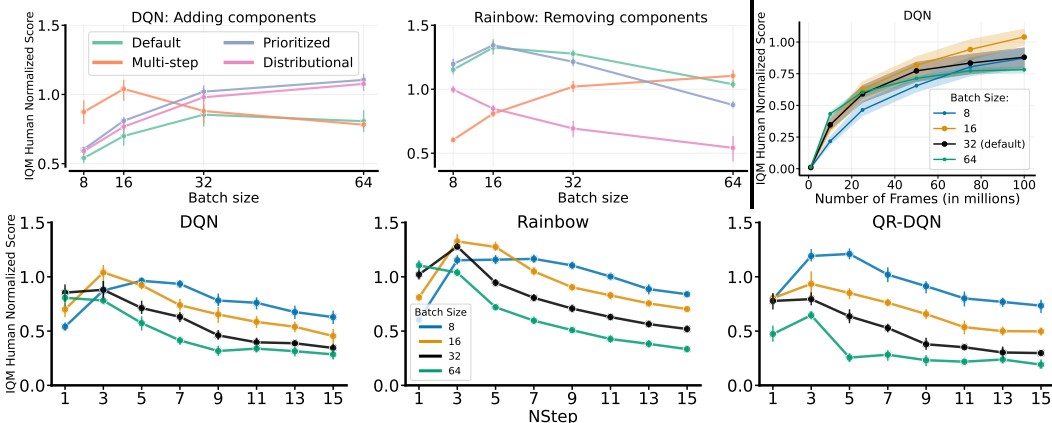

Figure 10: Measured IQM human normalized scores over 20 games with 3 independent seeds for each configuration, displaying 95% stratified bootstrap confidence intervals. **Top left:** Adding components to DQN; **Top center:** Removing components from Rainbow. **Top right:** Aggregate DQN performance with $n$-step of 3. **Bottom:** Varying batch sizes and $n$-steps in DQN (left), Rainbow (center), and QR-DQN (right).

Rainbow used is simply adding three components to the base DQN agent, we follow the analyses of Hessel et al. [2018] and Ceron and Castro [2021]. Specifically, in Figure 10 (top row) we simultaneously add these components to DQN (top left plot) and remove these components from Rainbow (top center plot). Remarkably, batch size is inversely correlated with performance *only when multi-step returns are used*. Given that DQN is the only agent considered here without multi-step learning, this finding explains the anomalous findings in Figure 3. Indeed, as the right panel of Figure 10 (top row) shows, adding multi-step learning to DQN results in improved performance with smaller batch sizes. To further investigate the relationship between batch size and multi-step returns, in Figure 10 (bottom row) we evaluate varying both batch sizes and $n$-step values for DQN, Rainbow, and QR-DQN. We can observe that smaller batch sizes suffer less from degrading performance as the $n$-step value is increased.

> **Key insights:**
> - The small batch effect does not seem to be a consequence of a sub-optimal choice of learning rate for the default value of 32.
> - The small batch effect does not arise due to beneficial interactions with the Adam optimizer.
> - The small batch effect appears to be more pronounced with multi-step learning.
> - When increasing the update horizon in multi-step learning, smaller batches produce better results.

## 4.2 Analysis of network optimization dynamics

In this section we will focus on three representative games (Asteroids, DemonAttack, and SpaceInvaders), and include results for more games in the supplemental material. In Figure 11 we present the training returns as well as a variety of metrics we collected for our analyses. We will discuss each in more detail below. The first column in this figure displays the training returns for each game, where we can observe the inverse correlation between batch size and performance.

**Variance of updates** Intuition suggests that as we decrease the batch size, we will observe an increase in the variance of our updates as our gradient estimates will be noisier. This is confirmed in the second column of Figure 11, where we see an increased variance with reduced batch size. A natural question is whether directly increasing variance results in improved performance, thereby (partially) explaining the results with reduced batch size. To investigate, we added Gaussian noise (at

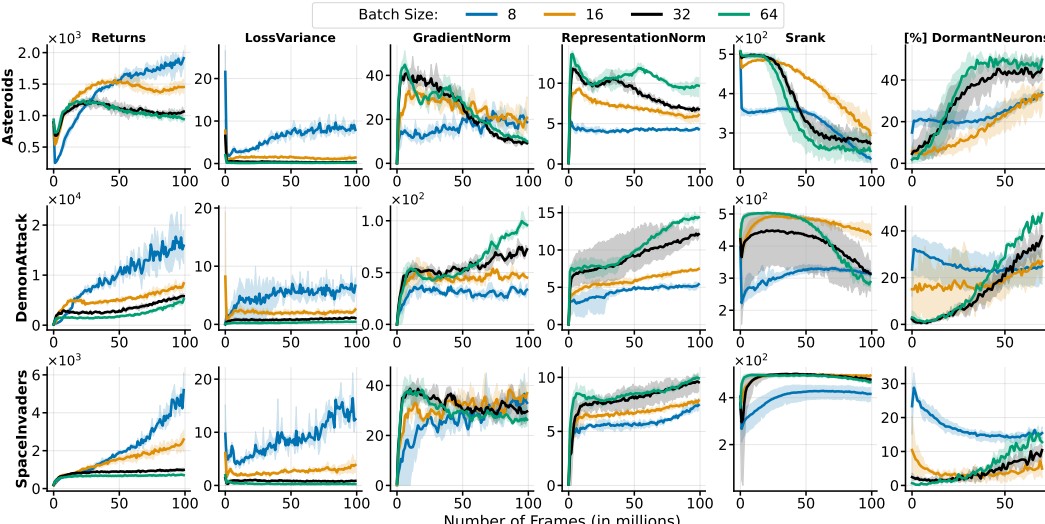

Figure 11: Empirical analyses for three representative games with varying batch sizes. From left to right: training returns, aggregate loss variance, average gradient norm, average representation norm, $srank$ [Kumar et al., 2021a], and dormant neurons [Sokar et al., 2023]. All results averaged over 3 seeds, shaded areas represent 95% confidence intervals.

varying scales) to the learning target $Q_{\bar{\theta}}$ (see section 2 for definition). As Figure 12 demonstrates, simply adding noise to the target does provide benefits, albeit with some variation across games.

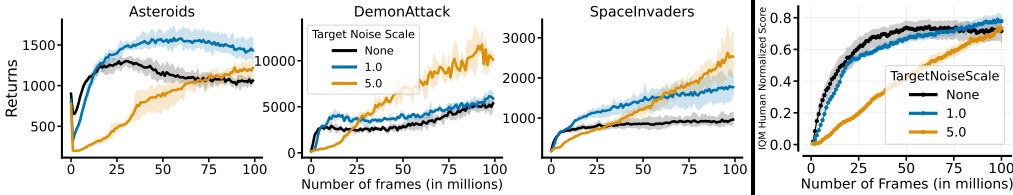

Figure 12: Adding noise of varying scales to the learning target with the default batch size of 32. **Left:** Performance of QR-DQN on three games with different target noise scale values. **Right:** Results aggregated IQM of human-normalized scores over 20 games for QR-DQN.

**Gradient and representation norms**    Keskar et al. [2017] and Zhao et al. [2022] both argue that smaller gradient norms can lead to improved generalization and performance, in part due to less "sharp" optimization landscapes. In Figure 11 (third column) we can see that batch size is, in fact, correlated with gradient norms, which may be an important factor in the improved performance. In Appendix D, we conducted experiments on a different subset of games, and observed a consistent trend: better performance is achieved with smaller batch sizes and gradient norms.

There have been a number of recent works suggesting RL representations, taken to be the output of the convolutional layers in our networks[6], yield better agent performance when their norms are smaller. Gogianu et al. [2021] demonstrated that normalizing representations yields improved agent performance as a result of a change to optimization dynamics; Kumar et al. [2021b] further observed that smaller representation norms can help mitigate feature co-adaptation, which can degrade agent performance in the offline setting. As Figure 11 (fourth column) shows, the norms of the representations are correlated with batch size, which aligns well with the works just mentioned.

**Effect on network expressivity and plasticity**    Kumar et al. [2021a] introduced the notion of the *effective rank* of the representation $srank_\delta(\phi)$[7], and argued that it is correlated with a network's expres-

---

[6]This is a common interpretation used recently, for example, by Castro et al. [2021], Gogianu et al. [2021], and Farebrother et al. [2023]

[7]$\delta$ is a threshold parameter. We used the same value of 0.01 as used by Kumar et al. [2021a].

sivity: a reduction in effective rank results in an implicit under-parameterization. The authors provide evidence that bootstrapping is the likeliest cause for effective rank collapse (and reduced performance).

Interestingly, in Figure 11 (fifth column) we see that with smaller batch sizes $srank$ collapse occurs earlier in training than with larger batch sizes. Given that there is mounting evidence that deep RL networks tend to overfit during training [Dabney et al., 2021, Nikishin et al., 2022, Sokar et al., 2023], it is possible that the network is better able to adapt to an earlier rank collapse than to a later one.

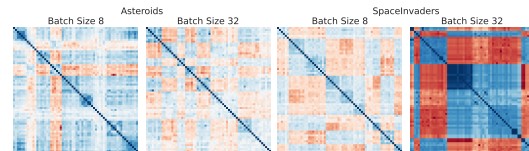

Figure 13: Gradient covariance matrices for Asteroids (**left**) and SpaceInvaders (**right**). In environments where smaller batch size significantly improves performance, it also induces weaker gradient correlation[8] and less gradient interference.

To further investigate the effects on network expressivity, we measured the fraction of *dormant neurons* (neurons with near-zero activations). Sokar et al. [2023] demonstrated that deep RL agents suffer from an increase in the number of dormant neurons in their network; further, the higher the level of dormant neurons, the worse the performance. In Figure 11 (rightmost column) we can see that, although the relationship with batch size is not as clear as with some of the other metrics, smaller batch sizes appear to have a much milder increase in their frequency. Further, there does appear to be a close relationship with the measured $srank$ findings above. Lyle et al. [2023] evaluated the covariance structure of the gradients to revisit the network's loss landscape, and argue that weaker gradient correlation and less gradient interference improve performance. We observe similar results in the gradient covariance heat maps shown in Figure 13 and Figure 16, where gradients appear to be largely colinear[8] when using larger batch size values.

> **Key insights:**
> - Reduced batch sizes result in increased variance of losses and gradients. This increased variance can have a beneficial effect during training.
> - Smaller batch sizes result in smaller gradient and representation norms, which tend to result in improved performance.
> - Smaller batch sizes seem to result in networks that are both more expressive and with greater plasticity.

## 5 Related work

There is a considerable amount of literature on understanding the effect of batch size in supervised learning settings. Keskar et al. [2016] presented quantitative experiments that support the view that large-batch methods tend to converge to sharp minimizers of the training and testing functions, and as has been shown in the optimization community, sharp minima tends to lead to poorer generalization. Masters and Luschi [2018] support the previous finding, presenting an empirical study of stochastic gradient descent's performance, and reviewing the underlying theoretical assumptions surrounding smaller batches. They conclude that using smaller batch sizes achieves the best training stability and generalization performance. Additionally, Golmant et al. [2018] reported that across a wide range of network architectures and problem domains, increasing the batch size yields no decrease in wall-clock time to convergence for either train or test loss.

Although batch size is central to deep reinforcement learning algorithms, it has not been extensively studied. One of the few results in this space is the work by Stooke and Abbeel [2018], where they argued that larger batch sizes can lead to improved performance when training in distributed settings. Our work finds the opposite effect: *smaller* batch sizes tends to improve performance; this suggests that empirical findings may not directly carry over between single-agent and distributed training scenarios. Islam et al. [2017] and Hilton et al. [2022] have investigated the role of batch size in on-policy algorithms. The latter demonstrates how to make these algorithms batch size-invariant, aiming to sustain training efficiency at small batch sizes.

---

[8] **Dark red** color refers to high negative correlation, and **dark blue** one high positive correlation.

Lahire et al. [2021] cast the replay buffer sampling problem as an importance sampling one, allowing it to perform well when using large batch. Fedus et al. [2020] presented a systematic and extensive analysis of experience replay in Q-learning methods, focusing on two fundamental properties: the replay capacity and the ratio of learning updates to experience collected (e.g. the replay ratio). Although their findings are complementary to ours, further investigation into the interplay of batch size and replay ratio is an interesting avenue for future work. Finally, there have been a number of recent works investigating network plasticity [Schwarzer et al., 2023, D'Oro et al., 2023, Sokar et al., 2023, Nikishin et al., 2022], but all have kept the batch size fixed.

Wołczyk and Krutsylo [2021] investigate the dynamics of experience replay in online continual learning, and focus on the effect of batch size choice when sampling from a replay buffer. They find that smaller batches are better at preventing forgetting than using larger batches, contrary to the intuitive assumption that it is better to recall more samples from the past to avoid forgetting. Additionally, the authors show that this phenomenon does not disappear under learning rate tuning. Their settings are similar to those used to generate Figure 3 in [Sokar et al., 2023], and suggest that target non-stationarity (e.g. bootstrapping) may have a role to play in explaining the small batch size effect we are observing.

## 6 Conclusions

In online deep RL, the amount of data sampled during each training step is crucial to an agent's learning effectiveness. Common intuition would lead one to believe that larger batches yield better estimates of the data distribution and yield computational savings due to data parallelism on GPUs. Our findings here suggest the opposite: the batch size parameter generally alters the agent's learning curves in surprising ways, and reducing the batch size below its standard value is often beneficial.

From a practical perspective, our experimental results make it clear that the effect of batch size on performance is substantially more complex than in supervised learning. Beyond the obvious performance and wall-time gains we observe, changing the batch size appears to have knock-on effects on exploration as well as asymptotic behaviour. Figure 8 hints at a complex relationship between learning rate and batch size, suggesting the potential usefulness of "scaling laws" for adjusting these parameters appropriately.

Conversely, our results also highlight a number of theoretically-unexplained effects in deep reinforcement learning. For example, one would naturally expect that decreasing the batch size should increase variance, and eventually affect prediction accuracy. That its effect on performance, both transient and asymptotic, should so critically depend on the degree to which bootstrapping occurs (as in $n$-step returns; Figure 10), suggests that gradient-based temporal-difference learning algorithms need a fundamentally different analysis from supervised learning methods.

**Future Work** Our focus in this paper has been on value-based online methods. This raises the question of whether our findings carry over to actor-critic methods, and different training scenarios such as offline RL [Levine et al., 2020] and distributed training [Stooke and Abbeel, 2018]. While similar findings are likely for actor-critic methods, the dynamics are sufficiently different in offline RL and in distributed training that it would likely require a different investigative and analytical approach. It is also an interesting direction to explore adaptive schemes that dynamically varies the batch size during training. Our experiments used a constant batch size, so further research is needed to determine whether it is advantageous to reduce the batch size over time in practice, as well as how quickly it should be reduced.

Our work has broader implications than just the choice of the batch size hyper-parameter. For instance, our findings on the impact of variance on performance suggest a promising avenue for new algorithmic innovations via the explicit injection of variance. Most exploration algorithms are designed for tabular settings and then adapted for deep networks; our results in section 3.5 suggest there may be opportunities for exploratory algorithms designed specifically for use with neural networks. We hope our analyses can prove useful for further advances in the development and understanding of deep networks for reinforcement learning.

**Acknowledgements.** Many thanks to Georg Ostrovski and Gopeshh Subbaraj for their feedback on an earlier draft of this paper. We also acknowledge Max Schwarzer, Adrien Ali Taiga, Rishabh Agarwal and Jesse Farebrother for useful discussions, as well as the rest of the DeepMind Montreal team for their feedback on this work. The authors would also like to thank the anonymous reviewers for useful feedback on this paper. Last but not least, we would also like to thank the Python community [Van Rossum and Drake Jr, 1995, Oliphant, 2007] for developing tools that enabled this work, including NumPy [Harris et al., 2020], Matplotlib [Hunter, 2007] and JAX [Bradbury et al., 2018].

**Broader impact** Although the work presented here is mostly of an academic nature, it aids in the development of more capable autonomous agents. While our contributions do not directly contribute to any negative societal impacts, we urge the community to consider these when building on our research

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

# A   Code availability

Our experiments were built on open source code, mostly from the Dopamine repository. The root directory for these is https://github.com/google/dopamine/tree/master/dopamine/, and we specify the subdirectories below (with clickable links):

- DQN, Rainbow, QR-DQN and IQN agents from /jax/agents/
- Atari-100k agents from /labs/atari-100k/
- Batch size from /jax/agents/quantile/configs/quantile.gin **(line 36)**
- Exploration $\epsilon = 0$ from /jax/agents/quantile/configs/quantile.gin **(line 16)**
- Resnet from /labs/offline-rl/jax/networks.py **(line 108)**
- Dormant neurons metric from /labs/redo/

For the srank metric experiments we used code from:
https://github.com/google-research/google-research/blob/master/
generalization_representations_rl_aistats22/coherence/coherence_compute.py

# B   Atari 2600 games used

Most of our experiments were run with 20 games from the ALE suite [Bellemare et al., 2013], as suggested by Fedus et al. [2020]. However, for the Atari 100k agents (subsection 3.3), we used the standard set of 26 games [Kaiser et al., 2020] to be consistent with the benchmark. Finally, we also ran some experiments with the full set of 60 games. The specific games are detailed below.

**20 game subset:** AirRaid, Asterix, Asteroids, Bowling, Breakout, DemonAttack, Freeway, Gravitar, Jamesbond, MontezumaRevenge, MsPacman, Pong, PrivateEye, Qbert, Seaquest, SpaceInvaders, Venture, WizardOfWor, YarsRevenge, Zaxxon.

**26 game subset:** Alien, Amidar, Assault, Asterix, BankHeist, BattleZone, Boxing, Breakout, ChopperCommand, CrazyClimber, DemonAttack, Freeway, Frostbite, Gopher, Hero, Jamesbond, Kangaroo, Krull, KungFuMaster, MsPacman, Pong, PrivateEye, Qbert, RoadRunner, Seaquest, UpNDown.

**60 game set:** The 26 games above in addition to: AirRaid, Asteroids, Atlantis, BeamRider, Berzerk, Bowling, Carnival, Centipede, DoubleDunk, ElevatorAction, Enduro, FishingDerby, Gravitar, IceHockey, JourneyEscape, MontezumaRevenge, NameThisGame, Phoenix, Pitfall, Pooyan, Riverraid, Robotank, Skiing, Solaris, SpaceInvaders, StarGunner, Tennis, TimePilot, Tutankham, Venture, VideoPinball, WizardOfWor, YarsRevenge, Zaxxon.

# C   Wall-time versus IQM of human-normalized

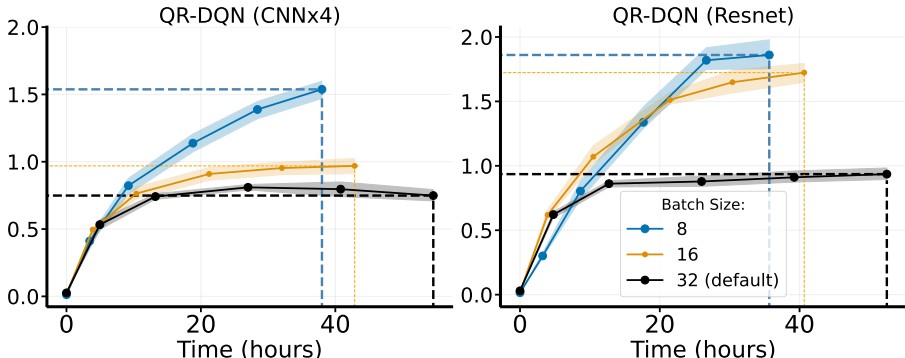

Figure 14: Measuring wall-time versus IQM of human-normalized scores when varying batch sizes and neural network architectures over 20 games in QR-DQN. Each experiment had 3 independent runs, and the confidence intervals show 95% confidence intervals.

# D    Average gradient norm

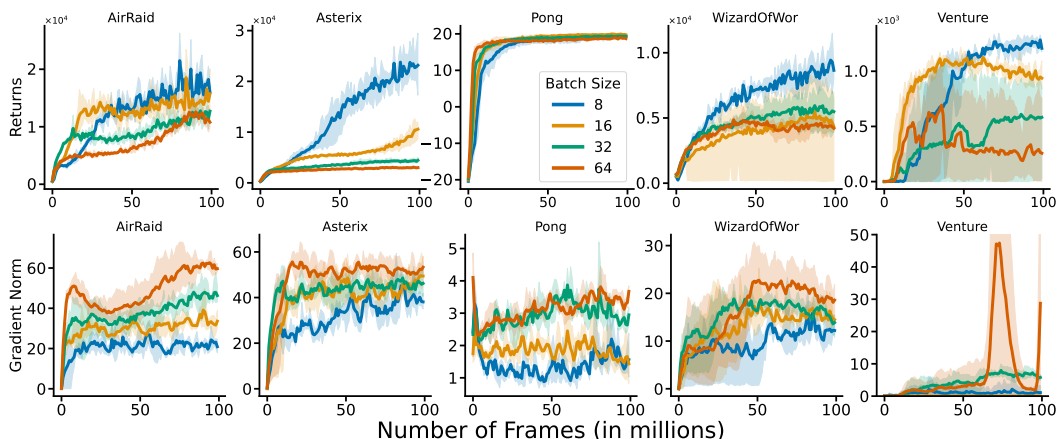

Figure 15: Empirical analyses for 5 representative games with varying batch sizes. **Top:** training returns, **Bottom:** average gradient norm. Results averaged over 3 seeds, shaded areas represent 95% confidence intervals.

# E    Gradient covariance

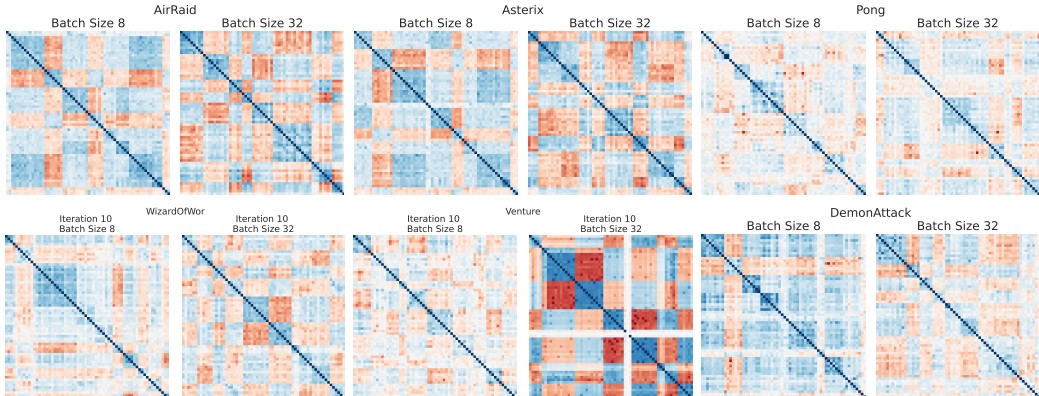

Figure 16: Gradient covariance plots for 6 representative games, which highlight the role of the gradient structure with varying batch sizes. We find that smaller batch size significantly improves performance and induces less gradient interference and weaker gradient correlation.

# F    Second order optimizer effects

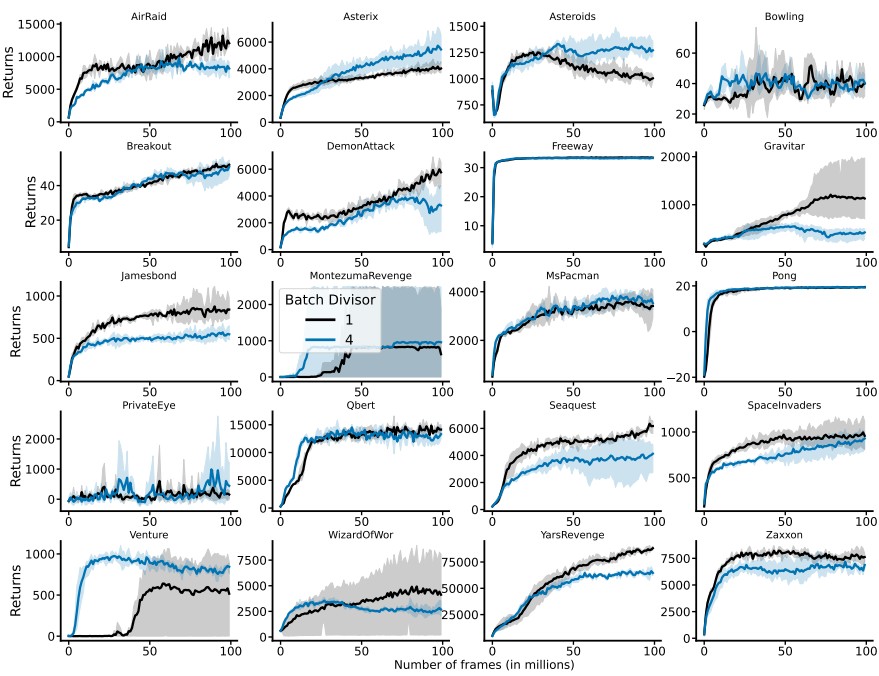

Figure 17: Evaluating multiple gradient updates per training step on QR-DQN, training curves for all games. Results averaged over 3 seeds, shaded areas represent 95% confidence intervals.

# G    Variance of updates.

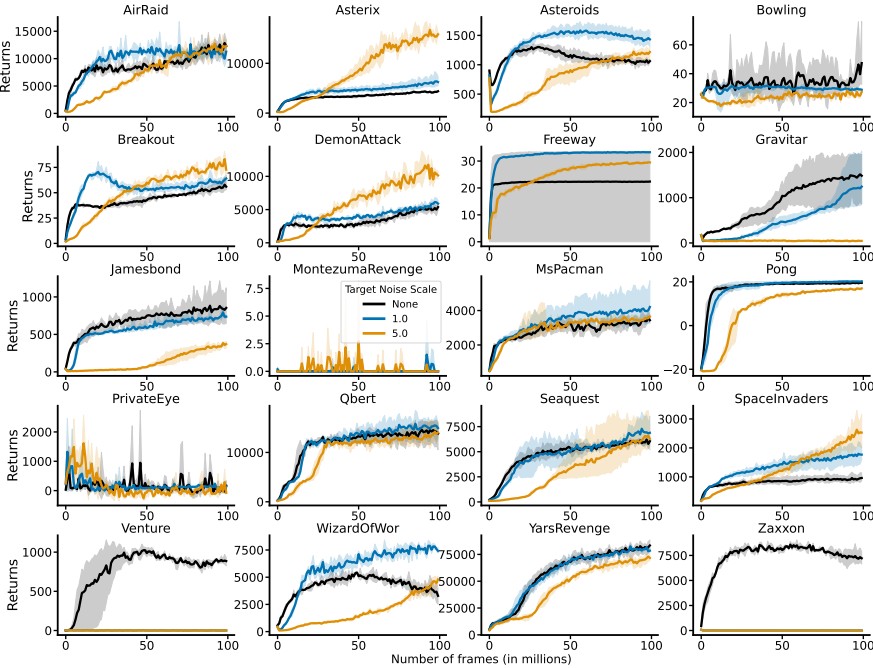

Figure 18: Evaluating the effect of adding target noise to QR-DQN, learning curves for all games. Results averaged over 3 seeds, shaded areas represent 95% confidence intervals.

# H    Results on the full ALE suite

We additionally provide complete results for all games using QR-DQN agent in Figure 19.

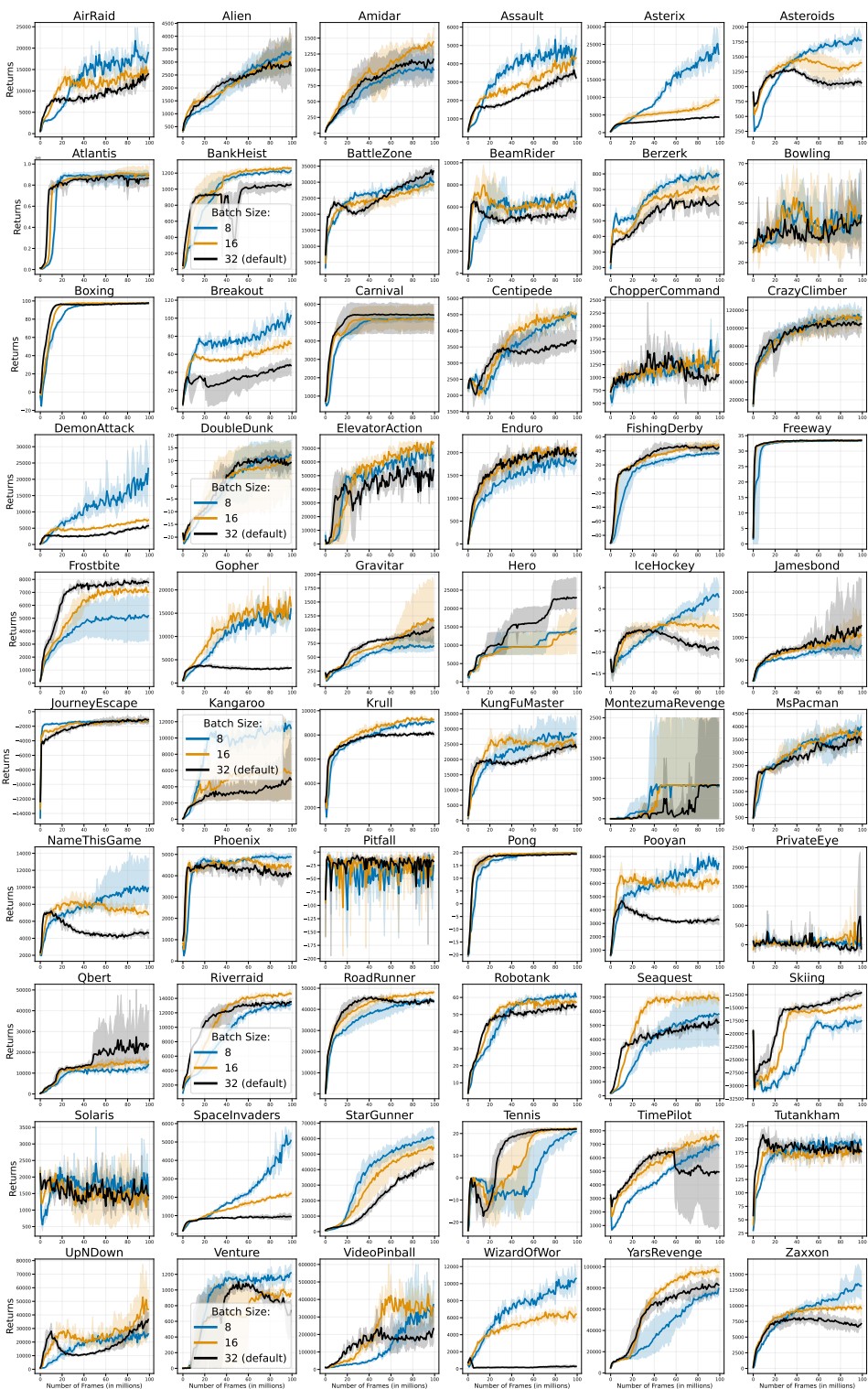

Figure 19: Training curves for QR-DQN agent. The results for all games are over 3 independent runs.

# I Varying architectures

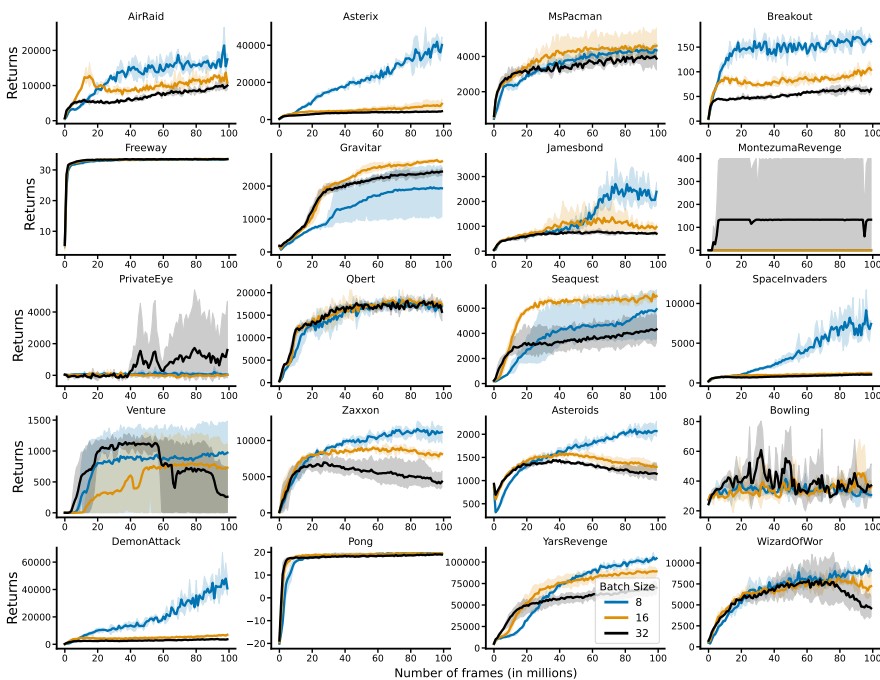

Figure 20: Evaluating the effect of CNNx4 to QR-DQN, learning curves for all games. Results averaged over 3 seeds, shaded areas represent 95% confidence intervals.

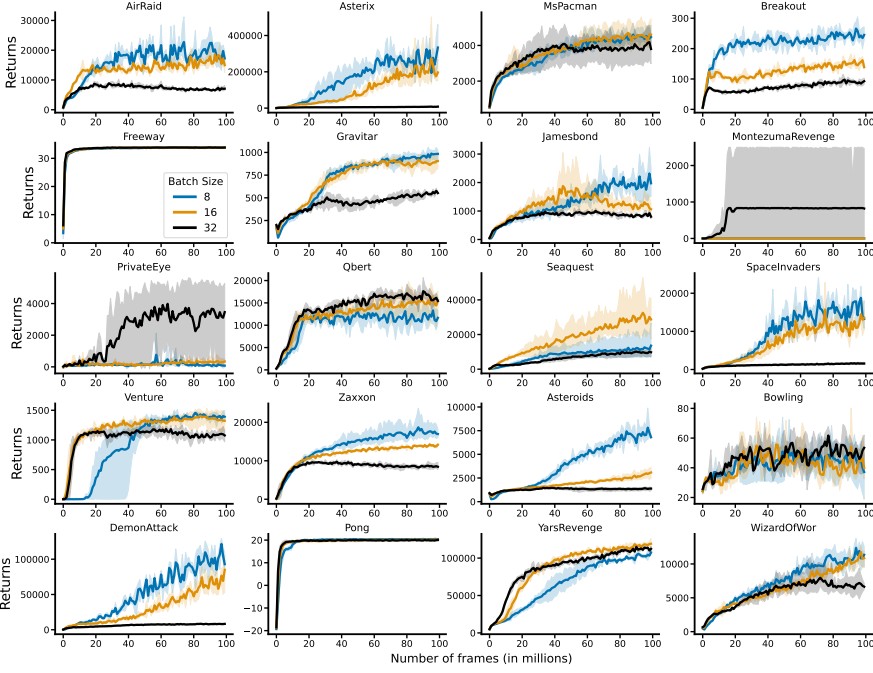

Figure 21: Evaluating the effect of Resnet to QR-DQN, learning curves for all games. Results averaged over 3 seeds, shaded areas represent 95% confidence intervals.

# J Training Stability

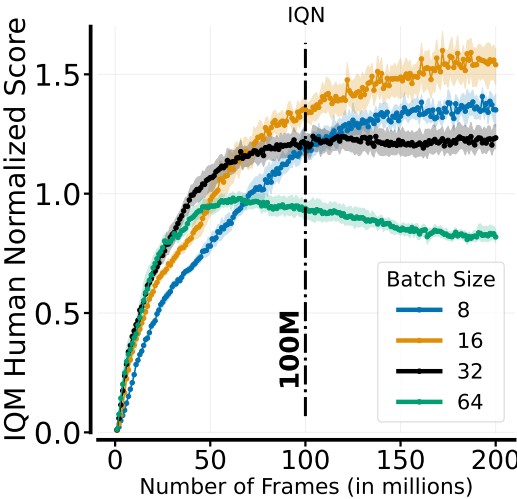

Figure 22: Measuring IQM for human-normalized scores when training for 200 million frames using IQN [Dabney et al., 2018b]. Results aggregated over 20 games, where each experiment was run with 3 independent seeds and we report 95% confidence intervals.

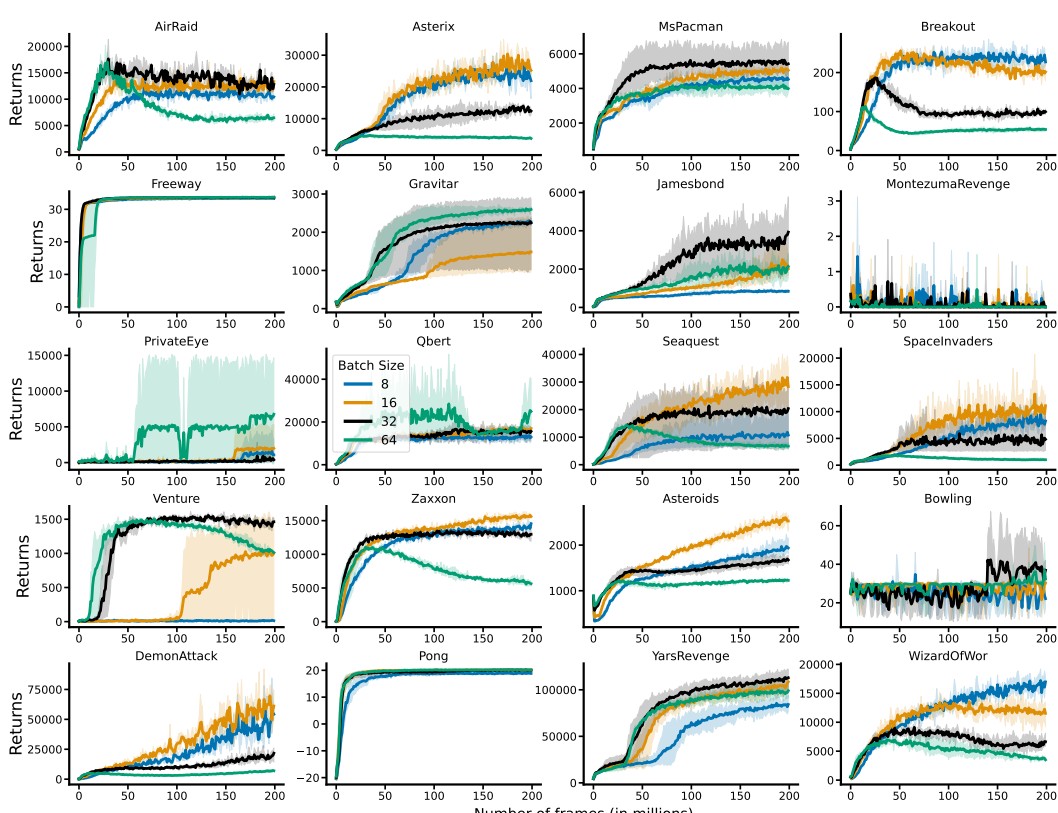

Figure 23: Learning curves for individual games, when trained for 200 million frames using IQN [Dabney et al., 2018b]. Results aggregated over 3 seeds, reporting 95% confidence intervals.

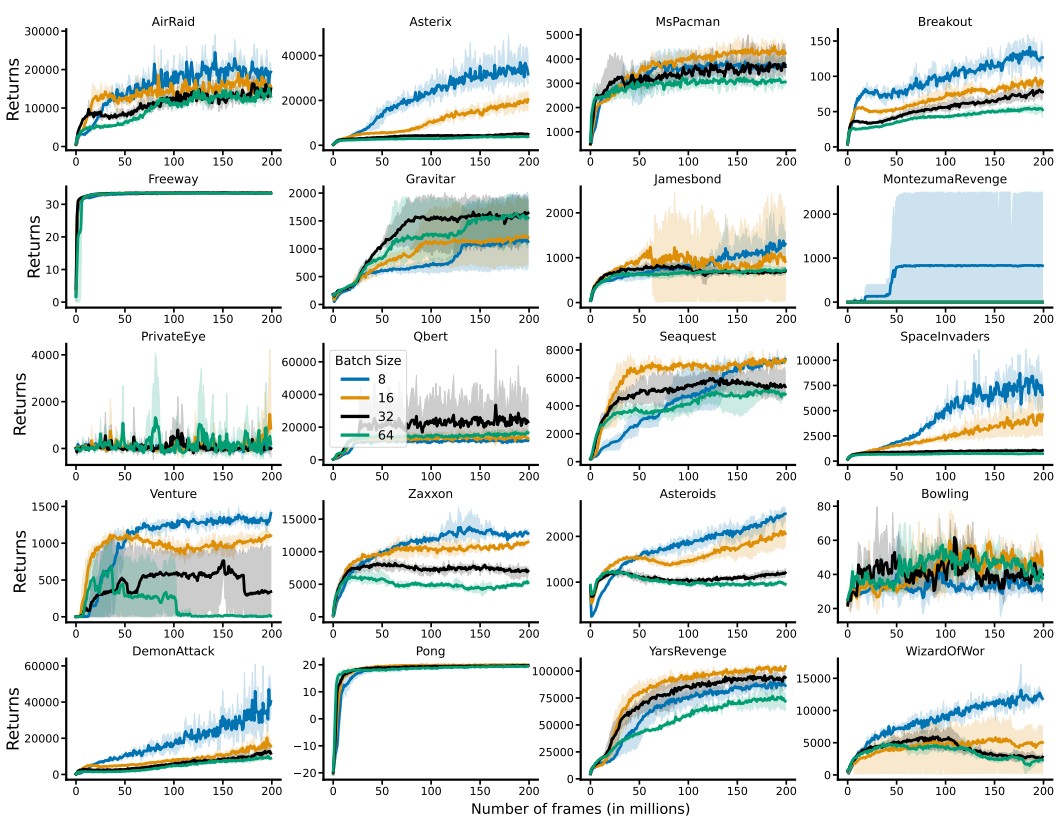

Figure 24: Learning curves for individual games, when trained for 200 million frames using QR-DQN [Dabney et al., 2018a]. Results aggregated over 3 seeds, reporting 95% confidence intervals.

