# OpenReview forum: "Small batch deep reinforcement learning"
_NeurIPS.cc/2023/Conference — NeurIPS 2023 poster_

### Official Review · Reviewer_vSto · 2023-06-09

**Soundness:** 4 excellent
**Presentation:** 4 excellent
**Contribution:** 3 good
**Rating:** 7
**Confidence:** 4

**Summary:**

The paper demonstrates the advantages of employing smaller batch sizes in value-based RL algorithms. It reveals that utilizing a smaller batch size can moderately or significantly enhance performance across several value-based RL algorithms, with the exception of DQN, where there are no improvements. Nonetheless, the paper reveals that incorporating a deeper network or employing n-step returns can restore the benefits of smaller batch sizes in the case of DQN.

To examine the underlying reasons behind the benefits of smaller batch sizes, the paper conducts a comprehensive set of experiments. These experiments investigate various factors, such as increased update variance, reduced representation norm, and improved network expressivity, shedding light on how smaller batch sizes can help performance.

**Strengths:**

- The paper presents a comprehensive and robust series of experiments, providing evidence for the effectiveness of smaller batch sizes in various value-based RL algorithms.
- The analysis conducted in the paper offers clear insights into the reasons why smaller batch sizes yield improvements in performance.
- The paper also delves into an in-depth investigation to understand why smaller batch sizes do not yield benefits in the case of DQN, providing valuable insights and explanations for this observed phenomenon.

**Weaknesses:**

While the result may not be technically novel, I agree that a comprehensive study on the topic of the benefits of smaller batch sizes is still highly valuable to the research community. Although some researchers are aware that smaller batch sizes can be beneficial, as seen in certain implementations of existing algorithms, such as IMPALA (e.g., https://github.com/facebookresearch/torchbeast uses a default batch size of 8), there is a lack of thorough analysis on this matter. Thus, the paper fills this gap and provides valuable insights and understanding of the issue. I appreciate the significance and value that the paper brings to the field.

**Questions:**

I suggest explaining the intuition on how the n-step return interplays with the batch size. My guess is the n-step return has a larger variance than a single-step return, and a smaller batch size can somehow further magnify this variance.

**Limitations:**

I don't see any clear limitations in this study.

---

> ### Author Rebuttal · Authors · 2023-08-09
>
> We thank the reviewer for the comments and appreciate the positive evaluation.
>
> Indeed, our intuition matches yours: the increase in variance from both fronts (smaller batch size and multi-step updates) seems to have a positive effect on performance. It is possible that they are adding different types of variance, which have a complementary effect on learning. This finding is further confirmed by Figure 2 in the PDF included with the general rebuttal at the top, which evaluates on classic control environments and shows that the small batch effect is also more pronounced when combined with multi-step updates.

---

> > ### Comment · Reviewer_vSto · 2023-08-10
> >
> > Thank you for your response. I suggest expanding more on the different types of variance in the revised paper and how the two variance interplay. My understanding is that the gradient update is the average of m iid random variables, and n-step return increases the variance by increasing individual variance, while a smaller batch size increases the variance by reducing m. This may explain how n-step return is needed for a small batch size to show effects, as the two sources of variance are multiplied together.
> >
> > I have no additional questions and will stand by my initial score.

---

> > > ### Author Response · Authors · 2023-08-18
> > > **Expanding discussion on different types of variance**
> > >
> > > Thank you for your suggestion. We agree with your interpretation of the two types of variance and agree that it would be worthwhile to expand on this in our revised paper!

---

### Official Review · Reviewer_scQb · 2023-07-04

**Soundness:** 4 excellent
**Presentation:** 4 excellent
**Contribution:** 3 good
**Rating:** 7
**Confidence:** 4

**Summary:**

This work studies the effect of reducing the batch size in value-based deep RL algorithms. Surprisingly, the authors find that smaller batch sizes generally improve learning performance and speed up training in terms of wall-clock time. Towards understand this "small batch effect", they empirically investigate how batch size relates to e.g. multistep learning, variance of gradient updates, network capacity, network plasticity, etc.

**Strengths:**

1. Overall, the writing is extremely clear and well-organized. While reading, I found myself asking questions that the authors then answered in later section of the paper (e.g. Lines 125-129 prompted questions on how batch size relates to plasticity and network capacity)
2. The empirical evaluation is thorough; the authors consider a wide range of Atari tasks and investigate how batch size relates to a variety of learning factors.
3. The relationships uncovered in this work relate to many RL research areas (e.g. exploration, continual learning), and I believe they will spur interesting future research.

**Weaknesses:**

1. The study only considers visual tasks with discrete actions. Does a small batch size improve data efficiency if you use the non-visual RAM observations in a few representative tasks? Do the same trends observed in Fig. 11 still hold? Since the paper is scoped to focused on value-based algorithms, I believe it is sufficient to state the discrete action limitation in the conclusion.
2. Since a smaller batch size seems to come with a variety of benefits (e.g. smaller gradient norms), it isn't clear to me if the observed benefits in Fig. 6 are due to improved exploration via higher variance gradient updates. Since it is likely not feasible to isolate exploration, can the authors instead clarify how these figures show improved exploration?

**Minor comments:**
2. Lines 59-60: "r" -> "r_t"
3. I believe line 265 should say "**decreasing** the batch size should increase variance"
3. Lines 266-268: "That it's effect..." This sentence is difficult to read and would benefit from rephrasing.

**Questions:**

1. Fig. 1: Can the authors provide any intuition on why 22/60 tasks performed worse with a smaller batch size with QR-DQN?
2. In Fig. 8, a larger batch size improves performance only for learning rate = 5e-06. Can the authors provide explanation?


**Limitations:**

See Weaknesses

---

> ### Author Rebuttal · Authors · 2023-08-09
>
> We thank the reviewer for the careful review of our paper, and suggestions for improvement. We address your concerns and questions below, referencing the PDF attached to the general response above. We will also correct all the minor issues pointed out in our submission.
>
> ## The study only considers visual tasks with discrete actions.
> Although our work focused on value-based algorithms, we agree it is worth investigating whether the effect is present in non-visual and continuous control tasks. We ran some experiments with DQN and Rainbow on 2 classic control environments (where inputs are state vectors) (Figure 2 in rebuttal PDF), as well as MPO [1] on DM-control environments (Figure 3 in rebuttal PDF). In both cases, we see a general trend towards improved performance when using smaller batches. In the classic control case we also see confirmation of the findings presented in section 4.1 of our submission: the small-batch effect is more pronounced with multi-step learning.
>
> ## Where is improved exploration coming from?
> Many methods have been proposed to address the exploitation-exploration dilemma, and some techniques emphasize exploration by adding noise directly to the parameter space of agents Fortunato et al., 2018 [2], Hao et al., 2023 [3], Plappert et al., 2017 [4], Gupta et al., 2018 [5], which inherently adds variance to the learning process. Noise perturbation is another approach that has been taken to induce exploration [6].
>
> Like these works, our analyses show that increasing variance by reducing the batch size may result in similar beneficial exploratory effects, as the mentioned works suggest. As the reviewer rightly points out, it is difficult to isolate the direct impact on exploration; however, the improved performance observed on all the hard exploration games in Atari, as well as in MountainCar (which is considered to be a hard exploration classic control environment, shown in Figure 2 in the rebuttal PDF) suggests that improved exploration may be an advantageous consequence of the variance induced by reduced batch size. As we discussed in section 5.1, we believe further work exploring the impact of variance injection in deep RL algorithms is necessary, and will add these points to our discussion.
>
> ## Why 22/60 of the games performed worse?
> It is very rare for an agent/algorithm to outperform the baselines on all games considered. For example, C51 [7] improved on 44/57, Rainbow [8] on 26 out of 57 games, QR-DQN [9] on 52/57, and Munchausen-IQN [10] on 40/60 games. Reporting aggregate performance, as is the norm with this benchmark, does mean we (as a community) gloss over some of these per-game differences. We agree this is an important problem the community should pay more attention to and will add a note to this effect in our discussion.
>
> ## Why larger batch size only improves for smaller learning rate?
> The default learning rate used (5e-05) was one optimized by prior work for a batch size of 32. It has been previously shown that one should reduce learning rates when increasing batch sizes [11], which is consistent with our findings (e.g. reducing the learning rate can be beneficial when increasing the batch size).
>
> # References
> *  [1] Abbas Abdolmaleki, Jost Tobias Springenberg, Yuval Tassa, Remi Munos, Nicolas Heess, and Martin Ried-miller. Maximum a posteriori policy optimisation. In International Conference on Learning Representations,2018.
> *  [2] Fortunato, M., Azar, M. G., Piot, B., Menick, J., Osband, I.,Graves, A., Mnih, V., Munos, R., Hassabis, D., Pietquin,O., Blundell, C., and Legg, S.  Noisy networks for exploration.  InProceedings of the International Confer-ence on Representation Learning (ICLR 2018), Vancou-ver (Canada), 2018.
> *  [3] Jianye Hao, Tianpei Yang, Hongyao Tang, Chenjia Bai, Jinyi Liu, Zhaopeng Meng, Peng Liu, and Zhen Wang.Exploration in deep reinforcement learning:  From single-agent to multiagent domain.IEEE Transactions on Neural Networks and Learning Systems, pages 1–21, 2023.
> *  [4] Plappert, M., Houthooft, R., Dhariwal, P., Sidor, S., Chen,R. Y., Chen, X., Asfour, T., Abbeel, P., and Andrychow-icz, M.   Parameter space noise for exploration. arXivpreprint arXiv:1706.01905, 2017
> *  [5] Abhishek  Gupta,  Russell  Mendonca,  YuXuan  Liu,  Pieter  Abbeel,  and  Sergey  Levine.   Meta-reinforcement learning of structured exploration strategies, 2018
> *  [6] Eberhard, O., Hollenstein, J., Pinneri, C., and Martius, G.Pink noise is all you need:  Colored noise exploration in deep reinforcement learning. InDeep ReinforcementLearning Workshop NeurIPS 2022, 2022
> *  [7] Marc G. Bellemare, Will Dabney, and Rémi Munos. A distributional perspective on reinforcement learning. In Proceedings of the 34th International Conference on Machine Learning - Volume 70, ICML’17, page 449–458, 2017.
> *  [8] Matteo Hessel, Joseph Modayil, Hado van Hasselt, Tom Schaul, Georg Ostrovski, Will Dabney, Dan Horgan, Bilal Piot, Mohammad Azar, and David Silver. Rainbow: Combining Improvements in Deep Reinforcement learning. In Proceedings of the AAAI Conference on Artificial Intelligence, 2018.
> *  [9] W. Dabney, M. Rowland, Marc G. Bellemare, and R. Munos. Distributional reinforcement learning with quantile regression. In AAAI, 2018.
> *  [10] Nino Vieillard, Olivier Pietquin, Matthieu Geist. Munchausen Reinforcement Learning. In Advances in Neural Information Processing Systems 33 (NeurIPS 2020).
> *  [11] D.Randall Wilson and Tony R. Martinez. The general inefficiency of batch training for gradient descent learning. Neural Networks, Volume 16, Issue 10, December 2003, Pages 1429-1451.

---

> > ### Comment · Reviewer_scQb · 2023-08-16
> >
> > Thank you for your response. All of my comments have been addressed, and I maintain my score.
> >
> > When I asked why smaller batch sizes decrease performance in 22/60 games, I should've been more specific. I meant to ask if these 22 games had anything in common that might explain why small batch sizes don't help? For instance, Fedus et. al [1] noted that most tasks saw an increase in performance when the replay buffer contained "younger" data (i.e data form more recent policies), though hard exploration tasks saw a significant drop in performance.
> >
> > [1] Revisiting Fundamentals of Experience Replay. Fedus et. al. ICML 2020.

---

> > > ### Author Response · Authors · 2023-08-18
> > > **Commonality between 22 games**
> > >
> > > Thank you for clarifying your point about the 22 games, which is a good question to raise. While we don't observe any clear game characteristic that would be indicative of whether it would benefit from smaller batch sizes, we do observe some commonality between some of the algorithms we considered. For example, in both YarsRevenge and JamesBond a batch size of 8 does the worst for QR-DQN (see Figure 15 in the appendix), which is also the case for IQN (see Figure 19 in the appendix).
> > >
> > > We are generating per-game plots for some of the other figures generated in the main paper and include them in the appendix, as these per-game results can prove useful for investigating commonalities across games, as you suggest. We will expand our discussion to include these points, thank you for suggesting it!

---

### Official Review · Reviewer_oeF6 · 2023-07-06

**Soundness:** 3 good
**Presentation:** 3 good
**Contribution:** 3 good
**Rating:** 7
**Confidence:** 5

**Summary:**

The work investigates the influence of replay batch size in experience replay for online reinforcement learning. The key finding is that reducing the batch size can be more beneficial, which contradicts common knowledge about regular deep learning.

**Strengths:**

The paper expands the analysis of an underinvestigated observation that can potentially change the default settings for experience replay and reduce the computational cost of further experiments. The authors provide new insights into the computational impact of batch size in experience replay and analyze network optimization dynamics. One of the strengths of the paper is its extensive experiments conducted with different settings and architectures.

**Weaknesses:**

Experience replay is also used in continual learning. It is surprising that the authors missed the paper that already drew the same conclusion [1], but kept it underinvestigated. Nonetheless, it should be mentioned in the related work section.

[1] Wołczyk, M., & Krutsylo, A. (2021). Remember More by Recalling Less: Investigating the Role of Batch Size in Continual Learning with Experience Replay (Student Abstract). AAAI Conference on Artificial Intelligence.

**Questions:**

-

---

> ### Author Rebuttal · Authors · 2023-08-09
>
> We thank the reviewer for their careful review of our submission, and for pointing us to the paper on experience replay in continual learning, it is indeed quite related.
>
> Wołczyk, M., & Krutsylo, A. (2021) investigate the dynamics of experience replay in online continual learning, and focus on the effect of batch size choice when sampling from a replay buffer. They find that smaller batches are better at preventing forgetting than using larger batches, contrary to the intuitive assumption that it is better to recall more samples from the past to avoid forgetting. Additionally, the authors show that this phenomenon does not disappear under learning rate tuning. Their settings are similar to those used to generate Figure 3 in Sokar et al., 2023 [1], and suggest that target non-stationarity (e.g. bootstrapping) may have a role to play in explaining the small batch size effect we are observing. We will add this to our discussion.
>
> # References
> [1] Ghada Sokar, Rishabh Agarwal, Pablo Samuel Castro, and Utku Evci. The dormant neuron phenomenon in deep reinforcement learning. In ICML, 2023.

---

> > ### Comment · Reviewer_oeF6 · 2023-08-13
> >
> > Thank you for your response. I am glad to see this improvement, and I have no further questions.

---

### Official Review · Reviewer_KbLx · 2023-07-07

**Soundness:** 2 fair
**Presentation:** 3 good
**Contribution:** 3 good
**Rating:** 4
**Confidence:** 4

**Summary:**

The authors study how batch size affects RL performance, and argue that a reduced batch size might (quite surprisingly) bring better performance improvement in a number of settings, in particular for QR-DQN, a smaller batch can lead to much better performance (almost doubling the performance). Different batch sizes are tested in Atari environments together with other hyperparameter changes. The authors also point out a benefit of using a smaller batch size is the reduction in wall clock time. The paper focus on empirical study and analysis, and provide a list of interesting findings.

**Strengths:**

**originality**
- the paper dedicates to bring a better understanding of effect of smaller batch sizes with extensive empirical studies, although there are already past works that study the effect of batch sizes, the results in this paper bring some new findings and observations and can be considered novel contribution.

**quality**
- overall presentation is good, but some arguments can be improved.
- extensive experiments and ablations are great

**clarity**
- Overall paper is clear to read. And the structure is easy to follow.

**significance**
- the observations presented in the paper can be interesting to the research community and help us better understand the effect of different batch size settings
- the authors argue that a smaller batch size can bring wall-clock time reduction and given the results they also have potential to bring better performance, which is a result that can be helpful towards better algorithms

**Weaknesses:**

Major concerns:

**related work**
- the authors claim "Surprisingly, to the best of our knowledge there have been no studies exploring the impact of the choice of batch size in deep RL." well, there are indeed some works that touch on this issue, for example to list a few:
  - Accelerated Methods for Deep Reinforcement Learning by Adam Stooke and Pieter Abbeel.
  - Reproducibility of Benchmarked Deep Reinforcement Learning Tasks for Continuous Control by Islam et al.
  - Shallow updates for deep reinforcement learning by Levine et al.
  - An Empirical Model of Large-Batch Training by McCandlish et al.
- these are older papers, please do some search on google scholar and have a better discussion of related works. Some of these works found larger batch size to be more beneficial. The authors need to spend a bit more effort in looking at related work and try to explain the discrepancy.


**Arguments and conclusions made in the paper**
- Some of the arguments can be improved, for example line 95: "In Figure 3 we can observe that, in general, reduced batch size results in improved performance." I don't think this is true, in Figure 3 there are 8 curves in 4 figures that have lower batch sizes than default, and 4 out of these 8 curves have weaker performance than the default, while the other 4 one can argue they are stronger or slightly stronger than default. One can argue in QR-DQN batch size of 8 is really good, but from this figure alone I don't see how it's a general trend.
- Line 226 Figure 11 (third column), I am not convinced there is a clear correlation between batch size and gradient norm. The variance is high, and for Asteroids and SpaceInvaders it seems in late stage of training they start to get higher gradient norms.
- Line 242 authors argue that " it is possible that the network is better able to adapt to an earlier rank collapse than to a later one." I am not sure how this argument is made, figure 11 col 5 shows that Srank does not positively correlate with performance at all. It seems this collapse has no negative effect on the training or even indicates stronger performance, which is again kind of going against what has been argued in previous literature. And the argument that "Smaller batch sizes seem to result in networks that are both more expressive and with greater plasticity." seems to be entirely wrong to me, as figure 11 shows on SpaceInvaders, small batch size of 8 has lowest srank and highest percentage of dormant neurons.
- Overall, I found a number of the points made in paper to be only supported by very weak evidence and they are not convincing. The authors might want to either modify the arguments into more accurate ones, or try to find better evidence to support the arguments. If the evidence is weak, the conclusion might not hold at all and it could be really coming from randomness or because of excessive fine-tuning on a particular algorithm on some particular environments.

The paper is currently lacking in these 2 aspects, but can be a good paper if these concerns are properly addressed.

Minor concerns:
- **originality** the novelty of the work is reduced by the fact that it is focused on studying existing methods, but mitigated by the novel empirical results, ablations and analysis.

**Questions:**

Suggestions as discussed in weakness section:
- spend more effort on related work and explain discrepancies from conclusions in the previous literature. After looking at these previous findings, why do you think your results are different from these findings that mostly say larger batch sizes are better?
- go through the arguments made in the paper, modify them to be more accurate or bring in better evidence to support them.

---

> ### Author Rebuttal · Authors · 2023-08-09
>
> We thank the reviewer for a careful read of our submission, and concrete suggestions for improving it. We address each point separately below, referencing the PDF attached to the general rebuttal at the top. We hope our responses, and the amendments we will make to our paper based on the points raised, are sufficient to address your concerns. Please let us know if you feel something was not properly addressed.
>
> ## Related work
> We thank the reviewer for pointing out these papers, which had escaped our attention (with the exception of S&A, which we were already citing). They are quite relevant to our work, so we will expand our discussion  to include them accordingly (specifics below). We will also soften the wording regarding no prior studies on batch size in RL.
>
> Some discussion points on the works suggested:
> * S&A focus on distributed training, which have very different learning dynamics than those considered in our work. Indeed, Stooke & Abeel emphasize _increasing_ batch size to obtain better performance.
> * Islam et al. focus on two policy-gradient methods (DDPG and TRPO), which again have quite different learning dynamics than the value-based methods we consider. Their findings seem to suggest that _larger_ batch sizes yield better performance, which contrasts with our findings.
> To investigate further, we ran some experiments on MPO [1] (an off-policy value-based method that still shares some of the benefits from on-policy policy-gradient methods like TRPO, see Figure 3 in rebuttal PDF). Our results suggest there is also a tendency towards improved performance with smaller batch sizes; we suspect this advantage is due to it being value-based, but more investigation is necessary. The evident differences on the effect of batch size on value-based versus policy-gradient methods are certainly worth discussing, and we will add a discussion to this effect, along with our new results.
> *  Levine et al. focus on "shallow" RL (e.g. using linear approximators instead of deep networks), and find that larger batch sizes yield better performance.
> *  McCandlish et al. seems to focus on very large batch sizes; the smallest batch size considered is 64, and they go as high as the millions for Dota 5v5. For their Atari results (most comparable to ours), the authors focus on A2C (another asynchronous policy gradient method) and find that the best batch sizes are 100-1000 early in training, and 400-8000 later in training, which are substantially larger than what we considered.
>
> Based on the reviewer suggestion, we will also be referencing and comparing our work to [2], which explores the connection between batch size and importance sampling, and [3,4] which discuss the impact of gradient norms on training (which our analyses show are connected to batch size choice).
>
> ## On 4/8 curves having weaker performance than default
> While true, it is worth noting that DQN in this figure does not have multi-step returns; as we discuss and analyze in section 4, we don’t observe improved performance with smaller batch sizes in DQN without multi-step returns. Indeed, as can be observed in Figure 10, both smaller-than-default batch sizes yield improved performance for DQN when multi-step updates are used. We will clarify this point in our discussion.
>
> ## Correlation between batch size & gradient norm
> For computational reasons, we focused on three games for our submission. However, we ran this analysis on five extra games (see Figure 1 in rebuttal PDF).  These show a stronger correlation between batch size and gradient norm. Nonetheless, we will modify our discussion to soften the claim, given your concerns, and include the extra figures in the appendix as further evidence of the connection.
>
> ## Connection between srank, plasticity, batch size, & performance
> The point we were making was based on the observation that in all three games we observed improved performance with reduced batch sizes, _and_ in all three games we observed an early collapse in srank. This is of course a correlation, and not necessarily a causal relationship, but we felt it was worth remarking on.
>
> The comment about plasticity was with regards to the discussion in the text: “although the relationship with batch size is not as clear as with some of the other metrics, smaller batch sizes appear to have a much milder increase in their frequency”. As Sokar et al. [5] showed, the level of dormant neurons increases (mostly) monotonically throughout training (as the black line for batch_size=32 shows); with smaller batch sizes we have a milder increase throughout training, although we do see them start with a higher fraction of dormant neurons.
>
> In both cases we can soften our claims and add the points made above to the discussion.
>
> ## Why are our findings different than those saying larger batch sizes are better?
> As discussed above, there appears to be a correlation between the use of distributed and/or policy gradient methods and improved performance due to larger batch sizes. Our work explores non-distributed value-based methods. As mentioned in the discussion above, we will highlight this point further in our discussion.
>
> # References
> *  [1] A. Abdolmaleki, J.T. Springenberg, Y. Tassa, R. Munos, N. Heess, and M. Riedmiller. Maximum a posteriori policy optimisation. ICLR 2018
> *  [2] T. Lahire, M. Geist, and E. Rachelson. "Large batch experience replay." arXiv preprint arXiv:2110.01528 (2021)
> *  [3] P. Mi, L. Shen, T. Ren, Y. Zhou, X. Sun, R. Ji, and D. Tao. "Make sharpness-aware minimization stronger: A sparsified perturbation approach." NeurIPS (2022)
> *  [4] X. Zhang, R. Xu, H. Yu, H. Zou, and P. Cui. "Gradient Norm Regularizer Seeks Flat Minima and Improves Generalization." (2022)
> *  [5] G. Sokar, R. Agarwal, P.S. Castro, and U. Evci. The dormant neuron phenomenon in deep reinforcement learning. In ICML, 2023

---

> > ### Author Response · Authors · 2023-08-18
> > **Are your concerns addressed?**
> >
> > Dear reviewer,
> > Given that the discussion period with authors is almost over, we wanted to reach out to see if there were any of your concerns you felt were not properly addressed, so that we may have time to respond to them if so. If all your concerns have been addressed, we would invite you to revise your score accordingly.
> >
> > Once again, thank you for the careful review of our paper!

---

> > > ### Comment · Reviewer_KbLx · 2023-08-21
> > > **Thank you for the rebuttal**
> > >
> > > I thank the authors for the rebuttal. There are definitely a lot of improvements, it's good to have some potential explanation for the discrepancy between this work and previous works, and it is great to make the arguments more accurate in some cases.
> > >
> > > I think the paper is quite interesting, but I'm still a little unsure whether the paper at this point is fully ready for publication, but I will take these into consideration in the reviewer discussion phase.

---

### Author Rebuttal · Authors · 2023-08-09

Dear reviewers and (S)ACs, we are attaching a PDF with three figures that we reference in each of our reviewer-specific rebuttals. The figures are:

**Figure 1:** Gradient variance analysis (with corresponding reward curves) for five extra Atari 2600 games, that help strengthen the claim of correlation between batch size, gradient variance, and agent performance. This is mostly in response to reviewer KbLx.

**Figure 2:** Effect of batch size on two classic control environments, which are state-based (as opposed to pixel-based, like Atari 2600 games). This is mostly in response to reviewer scQb.

**Figure 3:** Effect of batch size on MPO evaluated on DM-control. This is mostly in response to reviewers KbLx and scQb.

---

### Decision · Program_Chairs · 2023-09-21

**Decision:**

Accept (poster)

**Comment:**

This paper considers deep value-based RL in the small-batch regime, performing a controlled and in-depth study of observed benefits upon considering batch sizes smaller than those usually taken for granted in the literature, finding that variance is beneficial. The reviewers agree that the empirical study is quite high-quality, and that the phenomenon is interesting.

One reviewer raised concerns about the relationship with prior literature, which was amply addressed in the response.